# Dynamic Ensemble Modeling Approach to Nonstationary Neural Decoding in Brain-Computer Interfaces

**Yu Qi**[1], **Bin Liu**[2], **Yueming Wang**[3,*], **Gang Pan**[1,4,*]

qiyu@zju.edu.cn, bins@ieee.org, ymingwang@zju.edu.cn, gpan@zju.edu.cn

[1] College of Computer Science and Technology, Zhejiang University
[2] School of Computer Science, Nanjing University of Posts and Telecommunications
[3] Qiushi Academy for Advanced Studies, Zhejiang University
[4] State Key Lab of CAD&CG, Zhejiang University

## Abstract

Brain-computer interfaces (BCIs) have enabled prosthetic device control by decoding motor movements from neural activities. Neural signals recorded from cortex exhibit nonstationary property due to abrupt noises and neuroplastic changes in brain activities during motor control. Current state-of-the-art neural signal decoders such as Kalman filter assume fixed relationship between neural activities and motor movements, thus will fail if this assumption is not satisfied. We propose a dynamic ensemble modeling (DyEnsemble) approach that is capable of adapting to changes in neural signals by employing a proper combination of decoding functions. The DyEnsemble method firstly learns a set of diverse candidate models. Then, it dynamically selects and combines these models online according to Bayesian updating mechanism. Our method can mitigate the effect of noises and cope with different task behaviors by automatic model switching, thus gives more accurate predictions. Experiments with neural data demonstrate that the DyEnsemble method outperforms Kalman filters remarkably, and its advantage is more obvious with noisy signals.

## 1 Introduction

Brain-computer interfaces (BCIs) decode motor intentions directly from brain signals for external device control [1–3]. Intracortical BCIs (iBCIs) utilize neural signals recorded from implanted electrode arrays to extract information about movement intentions. Advances in iBCIs have enabled the development in control of prosthetic devices or computer cursors by neural activities [4].

In iBCI systems, neural decoding algorithm plays an important role. Many algorithms have been proposed to decode motor information from neural signals [5–7], including population vector [8], linear estimators [9], deep neural networks [10], and recursive Bayesian decoders [11]. Among these approaches, Kalman filter is considered to provide more accurate trajectory estimation by incorporating the process of trajectory evolution as a prior knowledge [12], which has been successfully applied to online cursor and prosthetic control, achieving the state-of-the-art performance [5, 13].

One critical challenge in neural decoding is the nonstationary property of neural signals [14]. Current iBCI neural decoders mostly assume a static functional relationship between neural signals and movements by using fixed decoding models. However, in an online decoding process, signals from neurons can be temporarily noised or even lost, and brain activities can also change due to

neuroplasticity [15]. With the presence of noises and changes, the functional mapping between neural signals and movements can be nonstationary and changes continuously in time [16]. Static decoders with fixed decoding functions can be inaccurate and unstable given nonstationary neural signals [14], thus need to be retrained periodically to maintain the performance [17, 18].

Most existing neural decoders dealing with nonstationary problems can be classified into two groups. The first group is recalibration-based, which uses a static model, and periodically recalibrates it (with offline paradigms) or adaptively updates the parameters online (usually requires true intention/trajectory). Most approaches belong to this group [5, 18, 19]. The second group uses dynamic models to track nonstationary changes in signals [20–22]. These approaches can avoid the expense of recalibration, which is potentially more suitable for long-term decoding tasks. However, there are very few studies in this group for the challenge in modeling nonstationary neural signals.

To obtain robust decoding performance with nonstationary neural signals, we improve upon the Kalman filter's measurement function by introducing a dynamic ensemble measurement model, called DyEnsemble, capable of adapting to changes in neural signals. Different from static models, DyEnsemble allows the measurement function to be adaptively adjusted online. DyEnsemble firstly learns a set of diverse candidate models. In online prediction, it adaptively adjusts the measurement function along with changes in neural signals by dynamically selecting and combining these models according to the Bayesian updating mechanism. Experimental results demonstrate that DyEnsemble model can effectively mitigate the effect of noises and cope with different task behaviors by automatic model switching, thus gives more accurate predictions. The main contributions of this work are summarized as follows:

- We propose a novel dynamic ensemble model (DyEnsemble) to cope with nonstationary neural signals. We propose to use the particle filter algorithm for recursive state estimation in DyEnsemble, which adaptively estimates the posterior probability of each candidate model according to incoming neural signals, and combines them online with the Bayesian updating mechanism. The process of dynamic ensemble modeling is illustrated in Fig. 1.

- We propose a candidate model generation strategy to learn a diverse candidate set from neural signals. The strategy includes a neuron dropout step to deal with noisy neurons, and a weight perturbation step to handle functional changes in neural signals.

Experiments are carried out with both simulation data and neural signal data. It is demonstrated that the DyEnsemble method outperforms Kalman filters remarkably, and its advantage is more obvious with noisy signals.

## 2   Dynamic Ensemble Modeling Algorithm

### 2.1   Classic state-space model

A classic state-space model consists of a state transition function $f(.)$ and a measurement function $h(.)$ as follows:

$$\mathbf{x}_k = f(\mathbf{x}_{k-1}) + \mathbf{v}_{k-1}, \tag{1}$$
$$\mathbf{y}_k = h(\mathbf{x}_k) + \mathbf{n}_k, \tag{2}$$

where $k$ denotes the discrete time step, $\mathbf{x}_k \in \mathbb{R}^{d_x}$ is the state of our interest, $\mathbf{y}_k \in \mathbb{R}^{d_y}$ is the measurement or observation, $\mathbf{v}_k$ and $\mathbf{n}_k$ are i.i.d. state transition noise and measurement noise.

In the context of neural decoding, the state and the measurement represent the movement trajectory and the neural signals, respectively. Given a sequence of neural signals $\mathbf{y}_{0:k}$, the task is to estimate the probability density of $\mathbf{x}_k$ recursively. For linear Gaussian cases, Kalman filter can provide an analytical optimal solution to the above task.

### 2.2   DyEnsemble based state-space model

In the classic state-space model mentioned earlier, the measurement function $h(.)$ is assumed to be precisely determined beforehand, which can not adapt to functional changes in neural signals. In DyEnsemble, we allow the measurement function to be adaptively adjusted online. Specifically, we

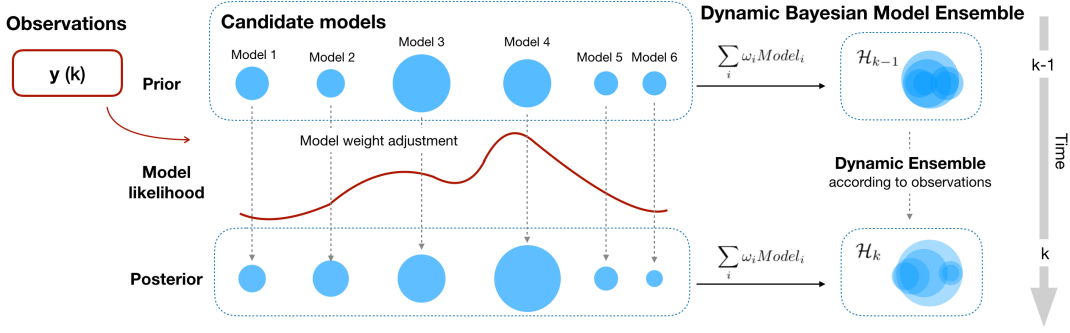

Figure 1: The dynamic ensemble modeling process.

consider an improved measurement model as follows:

$$\mathbf{y}_k = h_{\mathcal{H}_k}(\mathbf{x}_k) + \mathbf{n}_k, \tag{3}$$

in which $\mathcal{H}_k \in \{1, 2, \dots, M\}$ denotes the index of our hypotheses about the measurement function. Specifically, we use the notation $\mathcal{H}_k = m$ to denote the hypothesis that the working measurement function at time $k$ is $h_m$.

Here we adopt a set of functions, i.e. candidate models, to characterize the relationship between the measurement and the state to be estimated. A Bayesian updating mechanism [23–25] is used for dynamically switching among these models in a data-driven manner. Given an observation sequence $\mathbf{y}_{0:k}$, the posterior of the state at time $k$ is given by [26]:

$$p(\mathbf{x}_k|\mathbf{y}_{0:k}) = \sum_{m=1}^{M} p(\mathbf{x}_k|\mathcal{H}_k = m, \mathbf{y}_{0:k}) p(\mathcal{H}_k = m|\mathbf{y}_{0:k}), \tag{4}$$

where $p(\mathbf{x}_k|\mathcal{H}_k = m, \mathbf{y}_{0:k})$ is the posterior of the state corresponding to hypothesis $\mathcal{H}_k = m$; $p(\mathcal{H}_k = m|\mathbf{y}_{0:k})$ denotes the posterior probability of the $m$-th hypothesis.

Now we consider how to derive $p(\mathcal{H}_k = m|\mathbf{y}_{0:k})$ based on $p(\mathcal{H}_{k-1} = m|\mathbf{y}_{0:k-1})$. This is required for developing a recursive algorithm. Following [27], we specify a model transition process in term of forgetting, to predict the model indicator at $k$ as follows:

$$p(\mathcal{H}_k = m|\mathbf{y}_{0:k-1}) = \frac{p(\mathcal{H}_{k-1} = m|\mathbf{y}_{0:k-1})^\alpha}{\sum_{j=1}^{M} p(\mathcal{H}_{k-1} = j|\mathbf{y}_{0:k-1})^\alpha}, \tag{5}$$

where $\alpha$ ($0 < \alpha < 1$) denotes the forgetting factor which controls the rate of reducing the impact of historical data. Employing Bayes' rule, the posterior probability of the $m$-th hypothesis at $k$ is obtained as below:

$$p(\mathcal{H}_k = m|\mathbf{y}_{0:k}) = \frac{p(\mathcal{H}_k = m|\mathbf{y}_{0:k-1}) p_m(\mathbf{y}_k|\mathbf{y}_{0:k-1})}{\sum_{j=1}^{M} p(\mathcal{H}_k = j|\mathbf{y}_{0:k-1}) p_j(\mathbf{y}_k|\mathbf{y}_{0:k-1})}. \tag{6}$$

The term of $p_m(\mathbf{y}_k|\mathbf{y}_{0:k-1})$ is the marginal likelihood of model $h_m$ at time $k$, which is defined as:

$$p_m(\mathbf{y}_k|\mathbf{y}_{0:k-1}) = \int p_m(\mathbf{y}_k|\mathbf{x}_k) p(\mathbf{x}_k|\mathbf{y}_{0:k-1}) d\mathbf{x}_k, \tag{7}$$

where $p_m(\mathbf{y}_k|\mathbf{x}_k)$ is the likelihood function associated with the $m$-th hypothesis.

## 2.3 Particle filter algorithm for state estimation in DyEnsemble

Here we develop a generic particle-based solution to Eqn. (4) by adapting the particle filter (PF) to fit our model. In PF, the posterior distribution at each time step is approximated with a weighted particle set [28]. As shown in Eqn. (4), to estimate $p(\mathbf{x}_k|\mathbf{y}_{0:k})$ with particles, we need to derive a particle-based solution to: 1) $p(\mathbf{x}_k|\mathcal{H}_k = m, \mathbf{y}_{0:k})$; and 2) $p(\mathcal{H}_k = m|\mathbf{y}_{0:k})$.

Assume that we are standing at the beginning of the $k$-th time step, having at hand $p(\mathcal{H}_{k-1} = m|\mathbf{y}_{0:k-1}), m = 1, \ldots, M$, and a weighted particle set $\{\omega_{k-1}^i, \mathbf{x}_{k-1}^i\}_{i=1}^{N_s}$, where $N_s$ denotes the particle size, $\mathbf{x}_{k-1}^i$ the $i$-th particle with importance weight $\omega_{k-1}^i$. Assume that $p(\mathbf{x}_{k-1}|\mathbf{y}_{0:k-1}) \simeq \sum_{i=1}^{N_s} \omega_{k-1}^i \delta(\mathbf{x}_{k-1} - \mathbf{x}_{k-1}^i)$, where $\delta(.)$ denotes the Dirac delta function, we show how to get a particle solution to $p(\mathbf{x}_k|\mathcal{H}_k = m, \mathbf{y}_{0:k})$ and $p(\mathcal{H}_k = m|\mathbf{y}_{0:k})$, for $m = 1, \ldots, M$.

**Step 1. Particle based estimation of** $p(\mathbf{x}_k|\mathcal{H}_k = m, \mathbf{y}_{0:k})$. To begin with, we draw particles $\mathbf{x}_k^i$ from the state transition prior $p(\mathbf{x}_k|\mathbf{x}_{k-1}^i)$, for $i = 1, \ldots, N_s$. Then according to the principle of importance sampling, we have:

$$p(\mathbf{x}_k|\mathcal{H}_k = m, \mathbf{y}_{0:k}) \approx \sum_{i=1}^{N_s} \omega_{m,k}^i \delta(\mathbf{x}_k - \mathbf{x}_k^i), \tag{8}$$

where $\omega_{m,k}^i \propto \omega_{k-1}^i p_m(\mathbf{y}_k|\mathbf{x}_k^i)$, $\sum_{i=1}^{N_s} \omega_{m,k}^i = 1$. $\omega_{m,k}^i$ denotes the normalized importance weight of the $i$th particle under the hypothesis $\mathcal{H}_k = m$.

**Step 2. Particle based estimation of** $p(\mathcal{H}_k = m|\mathbf{y}_{0:k})$. Given $p(\mathcal{H}_{k-1} = m|\mathbf{y}_{0:k-1})$, first we calculate the predictive probability of $\mathcal{H}_k = m$ using Eqn. (5). Then we can calculate $p(\mathcal{H}_k = m|\mathbf{y}_{0:k})$ using Eqn. (6) provided that $p_m(\mathbf{y}_k|\mathbf{y}_{0:k-1}), m = 1, \ldots, M$ is available. Now we show how to make use of the weighted particle set in Step 1 to estimate $p_m(\mathbf{y}_k|\mathbf{y}_{0:k-1}), m = 1, \ldots, M$. Recall that, in Step 1, the state transition prior is adopted as the importance function, namely $q(\mathbf{x}_k|\mathbf{x}_{k-1}, \mathbf{y}_{0:k}) = p(\mathbf{x}_k|\mathbf{x}_{k-1})$. It naturally leads to a particle approximation to the predictive distribution of $x_k$, which is shown to be $p(\mathbf{x}_k|\mathbf{y}_{0:k-1}) \approx \sum_{i=1}^{N} \omega_{k-1}^i \delta_{\mathbf{x}_k^i}$. Then, according to Eqn. (7), we have

$$p_m(\mathbf{y}_k|\mathbf{y}_{0:k-1}) \approx \sum_{i=1}^{N_s} \omega_{k-1}^i p_m(\mathbf{y}_k|\mathbf{x}_k^i). \tag{9}$$

Note that PF usually suffers from the problem of particle degeneracy. That says, after several iterations, only a few particles have large weights. Hence, we adopt a resampling procedure in our method, which is a common practice in the literature for mitigating particle degeneracy by removing particles with negligible weights and duplicate particles with large weights.

## 2.4 Candidate model generation

Here we propose a candidate model generation strategy to learn a diverse model set from neural signals. The strategy includes two stages of neuron dropout and weight perturbation. To create proper candidate models, we analyze the properties of neural signals. The details of neural signal data are given in Section 4.1. The candidate model generation strategy is given in Algorithm 1.

**Neuron dropout.** Firstly, we evaluate the decoding ability of each neuron by mutual information (MI) between its spike rate and target trajectory in Fig. 2 (a). It shows that only some of the neurons contain useful information for motor decoding [29]. The activities of uncorrelated neurons can decrease decoding performance, and the informative neurons can also be temporarily noised or even lost. In neuron dropout, we randomly disconnect candidate models with several neurons to improve the noise-resistant ability and increase model diversity. After neuron dropout, each candidate only connects to a neuron subset containing $s$ neurons, where $s$ is the parameter of model size.

**Weight perturbation.** In Fig. 2 (b), we analyze the functional changes over time. Specifically, we fit the linear mapping function between neuron's firing rate and target trajectory in every 20-second temporal window with a stride of 1 second, and illustrate the distribution of the slope parameter. The red plus sign indicates the slope estimated with the whole time length. Results show that the mapping function between neuron and motor activity swings slightly around the mean in time.

To track the functional changes in neural signals, we propose a weight perturbation process. After model training, we randomly disturb the weights of each candidate model $h_m$ ($h_m \in \mathcal{M}$) in a small range:

$$w = w + p \times \epsilon, \tag{10}$$

where $\epsilon$ is randomly sampled from Gaussian(0,1), $p$ is the weight perturbation factor. The weight perturbation step gives the model set better tolerance of functional changes.

**Algorithm 1** Candidate Model Generation Strategy.

---

1: $s$: model size, $M$: model number, $p$: weight perturbation factor
2: $D$: training data, $\mathcal{N}$: neuron set
3: **Init** $\mathcal{M} = \{\}$
4: **for** $i = 1, ..., M$ **do**
5:      $N_{subset}$ = Neuron-dropout($N, s$)
6:      $h_i$ = Train-model($D, N_{subset}$)
7:      **for** $w$ in weights of $h_i$ **do**
8:          $w$ = Weight-perturbation($w, p$)
9:      **end for**
10:     Add $h_i$ to $\mathcal{M}$
11: **end for**

---

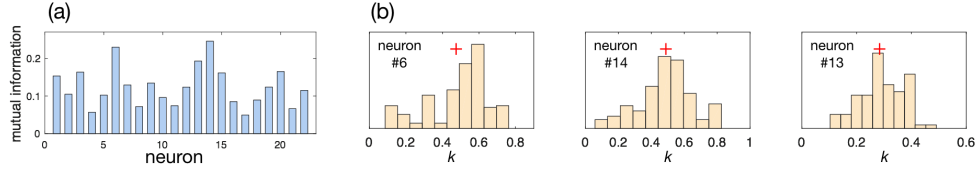

Figure 2: Neuron activity analysis.

## 3 Experiments with Simulation Data

The DyEnsemble approach is firstly evaluated with simulation data. In this experiment, we simulate a time series data where the measurement model is formulated by a piecewise function, to see how DyEnsemble tracks changes in functions.

The state transition function of the simulation data is given by:

$$x_{k+1} = 1 + sin(0.04\pi \times (k+1)) + 0.5x_k + v_k, \tag{11}$$

where $v_k$ is a Gamma(3,2) random state process noise. The formulation of state transition function follows [30]. The measurement function is defined as:

$$y_k = \begin{cases} h1(x) = 2x - 3 + n_k, & 0 < k \leqslant 100, \\ h2(x) = -x + 8 + n_k, & 100 < k \leqslant 200, \\ h3(x) = 0.5x + 5 + n_k, & 200 < k \leqslant 300, \end{cases} \tag{12}$$

where $n_k$ is Gaussian(0,1) random measurement noise. The goal is to estimate state $\mathbf{x}_k$ given a sequence of measurement $\mathbf{y}_{0:k}$. The length of simulation data is 300. In DyEnsemble, $h1$, $h2$, and $h3$ are adopted as candidate models. The forgetting factor $\alpha$ is set to 0.5, and the particle number is 200.

Fig. 3 shows the posterior probability of candidate models over time. In DyEnsemble, the weights of the candidate models switch automatically along with changes in signals. Candidate $h1$, $h2$, and $h3$ takes the dominating weight alternately, which is highly consistent with the piecewise function. We also evaluate the influence of forgetting factor $\alpha$, which adjusts the smoothness of weight transition. With a higher $\alpha$, model weights change more smoothly in time.

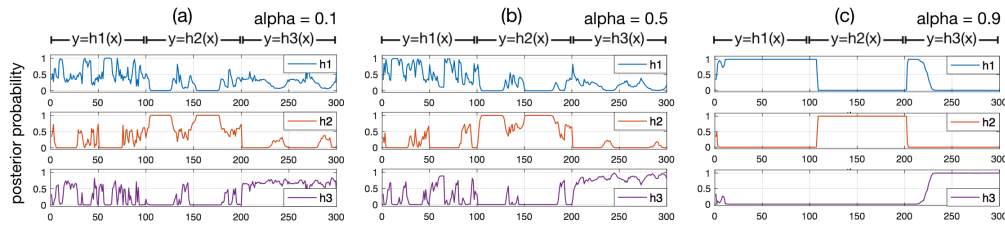

Figure 3: Weights of candidate models with different $\alpha$.

# 4 Experiments with Neural Signals

## 4.1 Neural signals and experiment settings

Neural signals were recorded from rats during lever-pressing tasks. The rats were trained to use their right forelimb to press a lever for water rewards. 16-channel microwire electrode arrays ($8\times2$, diameter = 35 $\mu m$) were implanted in the primary motor cortex of rats. Neural signals were recorded by a Cerebus$^{TM}$ system at a sampling rate of 30 kHz. Spikes were sorted and binned by 100 ms windows without overlap. The forelimb movements were acquired by lever trajectory, which was recorded at a sampling rate of 500 Hz, and downsampled to 10 Hz, to align to the spike bins. The experiments conformed to the Guide for the Care and Use of Laboratory Animals.

The neural signal dataset includes two rats, for each rat, the data length is about 400 seconds. We use the first 200 seconds for training and the last 100 seconds for test. After spike sorting, there are 22 and 58 neurons for rat1 and rat2 respectively. We evaluate the neurons with mutual information (MI) between the firing rate and lever trajectory, and select the top 20 neurons for movement decoding.

The movement trajectory at time step $k$ is a $3 \times 1$ vector $\mathbf{x}_k = [p_k, v_k, a_k]^T$, where $p_k$, $v_k$ and $a_k$ are the position, velocity and acceleration, respectively. The binned neural signal $\mathbf{y}_k$ is a $20 \times 1$ vector $\mathbf{y}_k = [y_k^1, y_k^2, ..., y_k^{20}]^T$, where each element $y_k^i$ denotes the spike count of the $i$-th neuron.

## 4.2 Analysis of dynamic process

**Adaptation to changing noises.** To analyze the dynamic ensemble process with changing noises, we add noise to neuron 2 and neuron 13 at around the 2nd and 4th second, as in Fig. 4 (a). The additional noise is randomly distributed integers in [0,10]. Fig. 4 (b) shows the weights of candidate models over time. There are a total of 20 candidate models with model size $s = 15$ and weight perturbation factor $p = 0.1$. Specially, we set the forgetting factor $\alpha = 0.98$ to force model weight transition to be highly smooth, for analysis convenience.

As shown in Fig. 4 (a) and (b), when there is no additional noise (the first 2 seconds), the 13th, 15th and 19th candidate models are with high weights in assembling, as in Fig. 4 (c). When noise occurs in the 2nd neuron (2-4 seconds), the weights of the 13th and 19th candidates become small because they both connect to the 2nd neuron. While the 15th candidate, which does not connect to neuron 2, takes the dominating weight, as in Fig. 4 (d). When noise occurs in neuron 13 at the 4th second, the weight of the 15th candidate decreases due to its connection to neuron 13, and the new winner is the 4th candidate, which does not connect to both noisy neurons (Fig. 4 (e)). The results strongly suggest that, given signals with changing noise, DyEnsemble approach can adaptively switch its model combination to mitigate the effect of noise.

**Adaptation to task behaviors.** Fig. 5 (a) visualizes the model weight transition process with $\alpha = 0.1$ and $\alpha = 0.5$. It is interesting to find that, the model assembling behaviors are different in lever-pressing and non-pressing periods. As highlighted in the dashed boxes in Fig. 5 (a), during lever-pressing, only several certain models are selected. In Fig. 5 (b) and (c), we illustrate the average weights of some candidate models in both lever-pressing and non-pressing periods. The results suggest that different models show different preferences to task behaviors, and DyEnsemble approach can automatically switch to suitable candidate models to cope with changes of behaviors.

## 4.3 Performance of neural decoding

Experiments are carried out to compare the neural decoding performance of DyEnsemble with other methods. The neural decoding performance is evaluated by commonly used criteria of correlation coefficient (CC) between lever trajectory and estimations. The results are shown in Table. 1.

To simulate noisy situations with unpredictable noises, we randomly replace several neurons' signals by noise in the test data. In Table. 1, we replace two (Noisy #2) and four (Noisy #4) neurons' signals by random integer noises in a range of [0, 10].

**Evaluation of neuron dropout and weight perturbation.** Here we evaluate the two key parts of neuron dropout and weight perturbation. In Table. 1, we compare the baseline approach (without neuron dropout and perturbation), the DyEnsemble with perturbation (p=0.1) alone, and the DyEnsemble with both perturbation (p=0.1) and dropout (DyEnsemble-2 and DyEnsemble-5 drop 2 and 5 neurons

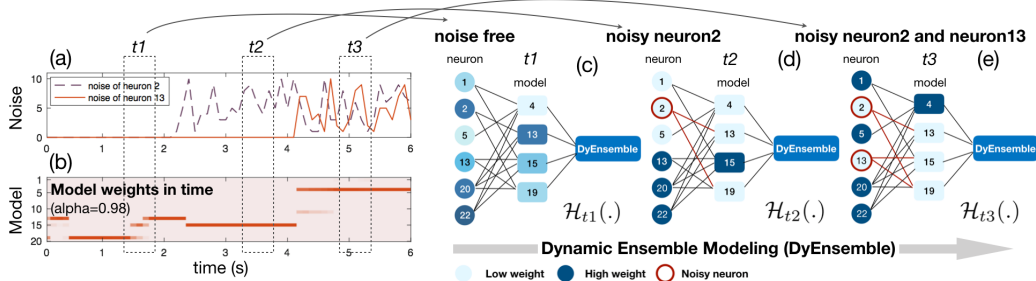

Figure 4: Dynamic ensemble modeling with changing noises.

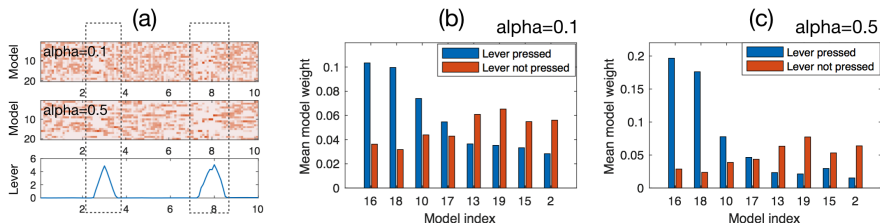

Figure 5: Comparison of model weights in lever-pressing and non-pressing periods.

respectively). Compared with the baseline, weight perturbation improves the performance by about 10% in noisy situations, and neuron dropout (DyEnsemble-5) leads to a further 20% performance improvement with 4 noisy neurons.

**Comparison with other decoders.** The methods in comparison include: Kalman filter, which is a baseline approach in neural motor decoding; long short-term memory (LSTM) [31] recurrent neural network, which excels in learning from sequential data in machine learning field [32]; dual decoder with a Kalman filter [21, 22], which can be regarded as the current state-of-the-art dynamic modeling approach for nonstationary neural signals.

For a fair comparison, we use linear functions of $f(.)$ and $h(.)$, zero-mean Gaussian terms of $\mathbf{v}_k$ and $\mathbf{n}_k$ in Kalman, dual decoder, and DyEnsemble. The $f(.)$ and $h(.)$ are estimated by the least square algorithm. For LSTM, we use a 1-hidden-layer model with 8 hidden neurons. In DyEnsemble, the forgetting factor $\alpha$ is 0.1, model number $M$ is 20, weight perturbation factor $p$ is 0.1, and the particle number is 1000. For DyEnsemble-2 and DyEnsemble-5, the model sizes are 18 and 15, respectively. All the methods are carefully tuned by validation and the validation set is the last 400 points of training data. The results are averaged over three random runs.

In Table. 1, we highlighted the top two performances in bold. Without additional noises (Original column), the CCs of DyEnsemble-2 and DyEnsemble-5 are 0.799 and 0.775 for Rat1, which are slightly higher than Kalman (0.777) and LSTM (0.753), and comparable to dual decoder (0.779). For Rat2, the best CC of DyEnsemble is 0.803 which is higher than Kalman (0.798) while slightly

Table 1: Correlation coefficient with different numbers of noisy neurons.

| Method | Rat 1 | | | Rat 2 | | |
|---|---|---|---|---|---|---|
| | Original | Noisy (#2) | Noisy (#4) | Original | Noisy (#2) | Noisy (#4) |
| Kalman filter | 0.777±0.000 | 0.696±0.012 | 0.560±0.009 | 0.798±0.000 | 0.580±0.039 | 0.381±0.093 |
| LSTM | 0.753±0.017 | 0.687±0.033 | **0.617±0.045** | **0.846±0.021** | 0.551±0.127 | 0.338±0.050 |
| Dual decoder | **0.779±0.000** | 0.694±0.010 | 0.575±0.013 | **0.803±0.000** | **0.585±0.025** | 0.387±0.030 |
| DyEnsemble (w/o P, w/o D) | 0.776±0.002 | 0.684±0.014 | 0.558±0.009 | 0.798±0.002 | 0.579±0.066 | 0.377±0.155 |
| DyEnsemble (P(0.1), w/o D) | 0.780±0.008 | 0.711±0.004 | 0.557±0.035 | 0.780±0.006 | 0.665±0.024 | 0.472±0.080 |
| DyEnsemble-2 (P(0.1), D(2)) | **0.799±0.012** | **0.735±0.006** | 0.583±0.090 | 0.788±0.009 | **0.633±0.064** | **0.516±0.092** |
| DyEnsemble-5 (P(0.1), D(5)) | 0.775±0.015 | **0.739±0.021** | **0.671±0.039** | **0.803±0.009** | 0.584±0.035 | **0.596±0.035** |

* w/o: without; P($k$): weight perturbation with p=$k$; D($l$): neuron dropout with $l$ neurons dropped.

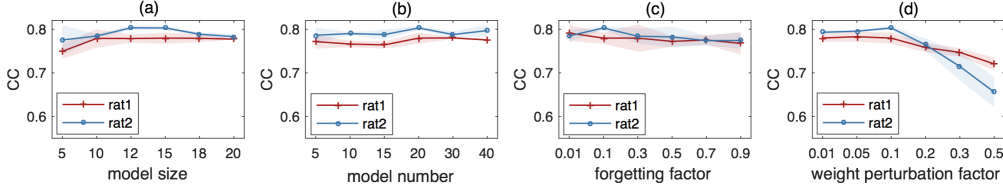

Figure 6: Evaluation of key parameters.

lower than LSTM (0.846). Overall, with original neural signals, the performance of DyEnsemble is comparable to state-of-the-art approaches.

With noisy neurons, the performances of Kalman, LSTM and dual decoder decrease significantly. For Rat1, when there are 2 noisy neurons, the CCs of Kalman, LSTM, and dual decoder are 0.696, 0.687 and 0.694, respectively. The CC of DyEnsemble-5 is 0.739, which improves by 6.2%, 7.6%, 6.5% compared with Kalman, LSTM, and dual decoder, respectively. With 4 noisy neurons, DyEnsemble-5 achieves a CC of 0.671, which is 19.8%, 8.8% and 16.7% higher than Kalman, LSTM and dual decoder, respectively. Similar results are observed with Rat2. DyEnsemble-5 is more stable and robust than DyEnsemble-2 with noisy neurons especially when there are 4 noisy neurons. Since Ensemble-2 only drops 2 neurons in candidate models, the performance decreases when more than 2 noisy neurons occur. Dropping more neurons can increase the robustness against noises, while it may also harm estimation accuracy.

### 4.4 Influence of parameters

Here we evaluate the key parameters in DyEnsemble: model size ($s$), model number ($M$), forgetting factor ($\alpha$), and weight perturbation factor ($p$). The results are illustrated in Fig. 6.

**Model size.** Model size $s$ defines the number of neurons connected to each candidate model. The rest of the parameters is set to: $\alpha = 0.1, p = 0.1, M = 20$. As shown in Fig. 6 (a), overall, CC improves with increase of model size. While for Rat2, CC decreases slightly after model size reaches 18. Commonly, a larger model size brings more information, while it also decreases the noise-resistant ability as discussed in Section 4.3.

**Model number.** The model number denotes the number of candidate models in $\mathcal{M}$. The rest of the parameters is set to: $\alpha = 0.1, p = 0.1, s = 15$. As shown in Fig. 6 (b), the performance increases when we tune $M$ from 5 to 20, while the improvement is subtle when $M$ is larger than 20.

**Forgetting factor.** The parameter of forgetting factor from Eqn. (5) controls the inertia in the candidate model transition. With a large $\alpha$, the candidate models prefer to keep the weights from the last time step. From Fig. 6 (c), we find that smaller $\alpha$ gives better performance in the neural decoding task, which reflects the nonstationary properties of neural signals. The rest of the parameters is set to: $s = 15, p = 0.1, M = 20$.

**Weight perturbation factor.** The weight perturbation factor $p$ controls the range that candidate models can deviate from the mean. A higher $p$ can tolerate larger changes in functional relationships, however, it also leads to inaccurate predictions. As shown in Fig. 6 (d), the performance improves when $p$ is adjusted from 0.01 to 0.1, while decreases rapidly when $p$ is bigger than 0.2. The rest of the parameters is set to: $s = 15, \alpha = 0.1, M = 20$.

## 5 Conclusion

We proposed a dynamic ensemble modeling approach, called DyEnsemble, for robust movement trajectory decoding from nonstationary neural signals. The DyEnsemble model improved upon the classic state-space model by introducing a dynamic ensemble measurement function, which is capable of adapting to changes in neural signals. Experimental results demonstrated that the DyEnsemble approach could automatically switch to suitable models to mitigate the effect of noises and cope with different task behaviors. The proposed method can provide valuable solutions for robust neural decoding tasks and nonstationary signal processing problems.

## 6 Acknowledgment

This work was partly supported by grants from the National Key Research and Development Program of China (2018YFA0701400, 2017YFB1002503), National Natural Science Foundation of China (61906166, 61571238, 61906099), Zhejiang Provincial Natural Science Foundation of China (LZ17F030001), and the Zhejiang Lab (2018EB0ZX01).

## Footnotes

*Corresponding authors: Yueming Wang and Gang Pan

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
