[Reviews · NeurIPS 2019]

Reviewer 1



Originality: - The paper references prior work and the authors note their approach differs from previous work that assumes fixed decoding models. Authors should include a brief summary of how motor imagery BCIs operate (note that there are a variety of BCI control signals, motor-imagery is one of them (line 17). - Authors modify a previously developed Bayesian algorithm to probabilistically weight candidate models and account for non-stationarities in EEG data. (Authors should revise their claim of “novelty” in main contributions, line 49 to state how they build on the previous Liu approach). Authors propose enhancements to the model by adapting model parameters based on tracking functional changes in neural signals and noisy neurons. Quality: - Overall, the technical content appears mostly correct, some information is missing. Sections 2.2 and 2.3 discuss previously developed algorithms/methods. o Line 86: Under what instances is low \alpha useful? o Line 97: What is w^i_{k-1} and how is it computed? o Authors should give range of parameters. - Technical details are missing in section 2.4. o What is the function form of the observation function, h_{ℋ_k}(x_k), and how are the models trained? o Authors should clarify if drop-out + perturbation steps are performed during model training or during online decoding. o Clarify the criterion used for neuron dropout. Authors state they “randomly disconnect model candidates” (line 127), but later state they “select 20 neurons with high MI values for movement decoding.” (lines 170-171). How is each neuron connected to a candidate model? Based on the information in Figure 2a, what neurons are dropped from the model? o Clarify how perturbation weights are updated. Using (10) or from data “based on the range that model candidates can deviate from the mean” (line 250-251). o Similar to what is provided in Algorithm 1, authors should include details on how decoding is performed (with steps described in 2.4). - Authors include results from simulated data and neural signal data to demonstrate the robust of their approach to dynamic changes by artificially inducing noise. Simulated experiment appears to be a replication of a previous study (Liu 2017) and does not demonstrate impact of model modifications. EEG experiments are well-designed. Performance is compared with various algorithms and results from empirical data indicate the weighted model is generally more robust under noisy conditions. A sensitivity analysis is also included to assess the impact of model parameters on performance. To demonstrate utility of the model enhancements, authors should include results without neuron dropout and/or perturbation. Clarity: - Overall, the paper is well-written and organised, and the technical content is well explained. A few areas need clarity (section 2.4 noted above). o Lines 43-45: “In online prediction, it adaptively adjusts the measurement function along with changes in neural signals by dynamically selects and combines these models according to Bayesian updating mechanism.” o Explain “spike sorting” (Line 168) o Use different nomenclature for model size (number of neurons/model) and model number (candidate models) as it can be confusing. o What does X in DyEnsemble(X) refer to? o Define notations earlier in text (e.g., “s” in line 179), instead of much later in section 4.4. o Table 1: Correlation coefficient between what? - Figure captions need to more descriptive so the reader does not rely on information embedded in the text to interpret the figures. Significance: - Paper is a useful contribution to the BCI community as it uses a weighted Bayesian model to address the problem of nonstationarities in neural signal data, in contrast to conventional methods that assume a fixed decoding model. Empirical results using EEG data demonstrate potential robustness to noisy conditions. - Algorithmic contributions are modifications to a previously developed Bayesian model. Although authors introduce modifications (neuron dropout and weight perturbation) to enhance model robustness, additional results are needed to assess the utility of the proposed enhancements. Post-rebuttal: Read authors' feedback and appreciated their responses to the main critiques, particularly novelty (slight modification (3) of Liu et al. (2017) to account for non-stationarities) and addition of neuron dropout (ND) + weight perturbation (WP), which should be stated in a revised manuscript. More impact if they would have demonstrated failure of Liu's approach with nonstationarities and a more robust simulation analysis (with realistic noise types). Authors' provided new results demonstrating utility of ND + WP in the neural experiment and comparison with a dual decoder. Based on feedback, revising initial review and score upward.

Reviewer 2



Originality: the methods per se are not new, but their combination is new, thus resulting in a new model. The authors describe that the state of the art is kalman filters to estimate trajectories from neural signals. KF is not a proper model for changing signals. However, I miss a specific state of the art description of models for non-stationary signals. I performed a very fast search of possible related papers and this topic has been previously addressed. Because these are not reported, comparison to other methods is not performed. Quality: i like the approach proposed by the authors because it can deal with noise as well as changing discriminative neurones. Their results support this claim. Clarity: this paper is clearly written. Significance: this approach is complete because it deals with noise and changing discriminative signals. Significance is difficult to evaluate due to lack of comparison to other techniques dealing with the same problem.

Reviewer 3



This is an exciting paper! The non-stationarity problem has plagued the iBCI field for a long time, and there's only been one other paper that I'm aware of that has attempted to address it in a non-heuristic way. Despite the shortcomings of this work, for the subcommunity that works with this type of data, this type of thinking is a big step forward, and I recommend publication. For the results, 20 neurons are available for decoding and DyEnsemble's candidates use either 15 or 18 neurons for decoding. With 18 neurons/candidate the model can ignore up to 2 noisy neurons, and with 15 neurons/candidate the model can ignore up to 5 noisy neurons. Unfortunately, baseline performance declines significantly when including fewer neurons per candidate, giving a tradeoff between noise robustness and baseline performance. Fortunately, I don't think this particular problem would pose as much of an issue in practice because modern electrode arrays have hundreds of channels or more - so even if we just take a subset of the neurons we would still have enough neurons for good performance in the large electrode arrays typical of clinical applications. Unfortunately, as the number of channels grow, the likelihood of having all noisy neurons excluded by a given randomly generated candidate model declines. So while this work improves robustness in a compelling way, there's still more work to be done. Update: After reviewing other reviewers' comments and the author feedback, am continuing to recommend acceptance.

[Author Response · NeurIPS 2019]

**1** **To Reviewer #1**

**2** **Novelty of our approach, compared with the previous approach (i.e. Liu 2017).** Our approach is NOT 'an
**3** application of the previous model (Liu 2017)'. Our proposed approach (i.e. DyEnsemble) consists of three main
**4** parts: state-space modeling, model candidate construction (a part of model-solving), and dynamic ensemble (a part of
**5** model-solving). 1) For the state-space modeling, we proposed a NOVEL dynamic observation formula (eq (3)), which
**6** described the nonstationary changes in neural signals. Liu's approach only work with multiple noise models, and could
**7** not describe changes of observation functions, thus it is unusable for nonstationary neural decoding. 2) For the model
**8** candidate construction, we proposed two new operations, namely Neuron dropout and Weight perturbation, to construct
**9** proper model candidates from neural signals. This stage is very critical for the effectiveness of our approach. 3) For
**10** dynamic ensemble, we mainly employed the framework of Liu's robust particle filter approach.

**11** **Effectiveness of Neuron dropout and Weight perturbation.** Indeed, we had evaluated the two new operations, and
**12** did not put in the paper for space limit. Part of them is shown in the table below. The results demonstrated that both
**13** dropout and perturbation significantly improve the correlation coefficient (CC) .The table includes the particle filter
**14** (PF) baseline (without neuron dropout and perturbation), the PF with perturbation (p=0.1) alone, and the PF with both
**15** perturbation (p=0.1) and dropout (drop 5 neurons). Compared with the PF baseline, weight perturbation improves
**16** the performance by about 10% in noisy situations. Neuron dropout operation leads to a further 20% performance
**17** improvement with 4 noisy neurons. Thanks.

Table 1: Evaluation of dropout and perturbation in terms of correlation coefficient (CC)

| Method | Rat 1 | | | Rat 2 | | |
|---|---|---|---|---|---|---|
| | Original | Noisy (#2) | Noisy (#4) | Original | Noisy (#2) | Noisy (#4) |
| PF Baseline | 0.776±0.002 | 0.684±0.014 | 0.558±0.009 | 0.798±0.002 | 0.579±0.066 | 0.377±0.155 |
| PF+Perturbation(0.1) | 0.780±0.008 | 0.711±0.004 | 0.557±0.035 | 0.780±0.006 | 0.665±0.024 | 0.472±0.080 |
| PF+Perturbation(0.1)+Dropout(5) | 0.775±0.015 | 0.739±0.021 | 0.671±0.039 | 0.803±0.009 | 0.584±0.035 | 0.596±0.035 |

**18** **Response to the questions.** (1) Under what instances is low $\alpha$ useful? - Low $\alpha$ can be useful when the adjacent time
**19** windows are not strongly correlated, e.g. with small time windows. (2) What is $w_{k-1}^i$ and how is it computed? - $w_{k-1}^i$
**20** is the weight of the $i$-th particle at time $k-1$. We initialize $w_0^i$ at time 0, and update it iteratively as described in Section
**21** 2.3. (3) What is the function form of the observation function, and how are the models trained. - The observation
**22** function takes form of $\mathbf{y} = \mathbf{A}\mathbf{x}$, and $\mathbf{A}$ is estimated by the least square algorithm. (4) For the other suggestions/issues,
**23** we will revise the paper accordingly. Many thanks for your valuable comments.

**24** **To Reviewer #2**

**25** **Description to techniques dealing with the same problem.** Most existing neural decoders dealing with nonstationary
**26** problem can be classified into two groups. The first group is recalibration-based, which uses a static model, and
**27** periodically recalibrates it (with offline paradigms) or adaptively updates the parameters online (usually require true
**28** intention/trajectory). Most approaches belong to this group (Gilja and Henderson 2015) (Shanechi and Carmena 2016).
**29** The second group uses dynamic models to track nonstationary changes in signals (Eden and Donoghue 2004) (Wang
**30** and Principe 2016). The dynamic model-based approaches can avoid the expense of recalibration, which are potentially
**31** more suitable for long-term decoding tasks. However, there is very few study in this group for the challenge to model
**32** nonstationary neural signals.

**33** **Comparison with state-of-the-art.** The proposed DyEnsemble approach belongs to the second group. Given strict time
**34** limit, we implemented dual decoder (Wang and Principe, 2016) with a Kalman filter, which tracks the gradual changes
**35** of individual neurons. The comparison with dual decoder is shown in the table below. Our approach demonstrates the
**36** superiority especially with noisy situations.

Table 2: Performance comparison in terms of correlation coefficient (CC)

| Method | Rat 1 | | | Rat 2 | | |
|---|---|---|---|---|---|---|
| | Original | Noisy (#2) | Noisy (#4) | Original | Noisy (#2) | Noisy (#4) |
| Dual decoder | 0.779±0.000 | 0.694±0.010 | 0.575±0.013 | **0.803±0.000** | 0.585±0.025 | 0.387±0.030 |
| DyEnsemble(18) | **0.799±0.012** | **0.735±0.006** | **0.583±0.090** | 0.788±0.009 | **0.633±0.064** | **0.516±0.092** |

**37** **To Reviewer #3**

**38** **About analysis of noises.** We injected noise into real data because it could provide an intuitive ground truth to
**39** investigate the dynamic ensemble process of candidate models. Indeed, analysis of real-world noises in neural signals
**40** would demonstrate stronger results. We are collecting some long-term neural signals to analyze real-world noises. For
**41** the baseline approach you mentioned, we will add discussions to compare with it. Thanks for the thoughtful review and
**42** constructive suggestions.

[Meta-Review · NeurIPS 2019]

This paper is on dynamic neural decoding via a state space model which ensembles linear dynamical systems. The specification of the ensemble is creative, but, methodologically, the rest of the paper is straightforward. Nevertheless, reviewers found that this paper has tackled an important application in a sound way and is a step forward for the community interested in this area. Please consider carefully the improvement suggestions provided by the reviewers.